# A Tightly Coupled Visual-Inertial GNSS State Estimator Based on Point-Line Feature

**DOI:** 10.3390/s22093391

**Published:** 2022-04-28

**Authors:** Bo Dong, Kai Zhang

**Affiliations:** 1Tsinghua Shenzhen International Graduate School, Tsinghua University, Shenzhen 518055, China; db19@mails.tsinghua.edu.cn; 2Research Institute of Tsinghua, Pearl River Delta, Guangzhou 510530, China

**Keywords:** GNSS-VIO, line feature, carrier phase smoothed pseudorange, parameter calibration, observability

## Abstract

Visual-inertial odometry (VIO) is known to suffer from drifting and can only provide local coordinates. In this paper, we propose a tightly coupled GNSS-VIO system based on point-line features for robust and drift-free state estimation. Feature-based methods are not robust in complex areas such as weak or repeated textures. To deal with this problem, line features with more environmental structure information can be extracted. In addition, to eliminate the accumulated drift of VIO, we tightly fused the GNSS measurement with visual and inertial information. The GNSS pseudorange measurements are real-time and unambiguous but experience large errors. The GNSS carrier phase measurements can achieve centimeter-level positioning accuracy, but the solution to the whole-cycle ambiguity is complex and time-consuming, which degrades the real-time performance of a state estimator. To combine the advantages of the two measurements, we use the carrier phase smoothed pseudorange instead of pseudorange to perform state estimation. Furthermore, the existence of the GNSS receiver and IMU also makes the extrinsic parameter calibration crucial. Our proposed system can calibrate the extrinsic translation parameter between the GNSS receiver and IMU in real-time. Finally, we show that the states represented in the ECEF frame are fully observable, and the tightly coupled GNSS-VIO state estimator is consistent. We conducted experiments on public datasets. The experimental results demonstrate that the positioning precision of our system is improved and the system is robust and real-time.

## 1. Introduction

Localization plays an important role in many applications such as robotics, unmanned driving, and unmanned aerial vehicles (UAVs). The fusion of information from multiple sensors for localization is the current mainstream method. The sensor fusion approaches have been widely studied for decades and can be divided into two streams: loosely coupled and tightly coupled approaches. Loosely coupled approaches [1,2] process the measurements of each sensor individually and then fuse all the results to obtain the final results. Tightly coupled approaches [3,4] combine measurements from all sensors and then use an estimator to process these measurements to obtain the final results. In general, the accuracy of tightly coupled approaches will be higher than that of loosely coupled approaches.

Visual odometry (VO) has received more attention due to the low price of cameras. Davison et al. [5] introduced MonoSLAM, which is the first monocular visual SLAM system that extracts the Shi-Tomasi corner [6]. MonoSLAM actively searches for matching point features in projected ellipses. However, because of the high computational complexity of MonoSLAM, it can only handle some small scenes. Klein et al. [7] proposed PTAM, which is the first optimization-based monocular visual SLAM system that executes feature tracking and mapping as two independent tasks in parallel in two threads. 

Because PTAM is designed for small-scale AR scenes, it also has some disadvantages. For example, it can only handle small-scale scenes, and the tracked features are easy to lose. A milestone work after PTAM is ORB-SLAM [8] and ORB-SLAM2 [9]. ORB-SLAM improves PTAM by detecting and tracking ORB features and adding loop closure. These improvements greatly improve the positioning precision of VO. However, ORB-SLAM can only build sparse point cloud maps and cannot track keyframes during pure rotation, which also limits its application. 

In addition, when encountering environments with repeated textures, point features cannot express environmental structure information well. On the contrary, line features contain more information related to the environmental structure and can overcome the interference caused by repeated textures. Therefore, point-line feature fusion can ensure that the VO system is more robust. He Yijia et al. [10] proposed PL-VIO, which is a VIO system based on point-line features. It uses the LSD line segment extraction algorithm [11] from OpenCV [12] to detect line features. However, LSD becomes the bottleneck for real-time performance due to its high computational cost [13], which causes the performance of PL-VIO to be seriously affected. To improve the speed and positioning precision of PL-VIO, Fu Qiang et al. [14] proposed PL-VINS, which is also a VIO framework based on point and line features. PL-VINS designs a modified LSD algorithm by studying hidden parameter tuning and length rejection strategy. The modified LSD runs at least three times faster than the LSD. PLF-VINS [15] is another VIO system based on point-line features, which introduces two methods for fusing point and line features. The first method is to use the positional similarity of point-line features to search for the relationship between point and line features. The second method is to fuse 3D parallel lines. The residuals formed by the two methods are then added to the VIO system. The results of PLF-VINS show that its positioning precision is greatly improved compared to some classic SLAM systems, such as OKVIS [4] and VINS-Mono [16].

Line features can improve the positioning precision of the VO system. However, in weak textures or dark scenes, the system still cannot extract a sufficient number of point and line features, which can cause large positioning errors. To overcome the shortcoming, fusing IMU measurements with images is a very effective method. MSCKF [17,18] is a VIO system based on EKF, which adds camera poses to the states of the system. When cameras observe landmarks, constraints can be formed between camera poses. The system is then updated by an observation model derived from geometric constraints. Since the number of camera poses is much smaller than the number of landmarks, this greatly reduces the time complexity of the system. VINS-Mono [16] is a tightly coupled, nonlinear optimization-based method that can obtain high-precision positioning results. Due to the existence of the loop closure and 4-DoF pose graph optimization, even if the system runs in large-scale scenes, it can still obtain accurate positioning results. In addition, to improve the performance of ORB-SLAM and eliminate the errors of ORB-SLAM, ORB-SLAM3 [19] adds IMU measurements based on ORB-SLAM. ORB-SLAM3 is more robust and has higher positioning precision compared with ORB-SLAM. 

The VIO system is more robust than the VO system. Nevertheless, the VIO system has four unobservable directions, namely *x*, *y*, *z*, and *yaw*, which lead to the accumulated drift of VIO. To eliminate the accumulated drift, an effective approach is to combine VIO with GNSS measurements. Lee et al. [20] demonstrated that the GPS-aided VIO system is fully observable in the ENU frame. GVINS [21] is a tightly coupled GNSS-VIO state estimator that combines VIO with GNSS pseudorange and Doppler frequency shift measurements, which achieves high positioning accuracy and is a state-of-the-art method. Li Xingxing et al. [22] introduced a semi-tightly coupled framework based on GNSS Precise Point Positioning (PPP) and stereo VINS. The system can use S-VINS to return accurately predicted positions in GNSS-unfriendly areas. Liu Jinxu et al. [23] also proposed a tightly coupled GNSS-VIO state estimator that fuses GNSS raw measurements with VIO. It drops all GNSS measurements in GNSS-degradation scenes, which limits its positioning precision.

However, there are many challenges in fusing the information from multiple sensors. First, the VIO system cannot extract enough point features in areas with repeated textures. Second, GNSS Single Point Positioning (SPP) uses pseudorange measurement; it can only achieve meter-level positioning accuracy. Therefore, if the pseudorange measurement and VIO are directly fused, the positioning accuracy is not greatly improved. Finally, since the GNSS receiver and IMU are fixed in different spatial positions, it is necessary to estimate the extrinsic parameter between them.

In response to the above problems, this paper proposes a new tightly coupled GNSS-VIO system. Our proposed system is drift-free and can provide global coordinates. The main contributions of the text are as follows:To obtain environmental structure information and deal with environments with repeated textures, we extracted line features based on point features.To combine the merits of the pseudorange and carrier phase measurements, we used the carrier phase smoothed pseudorange instead of the pseudorange measurement, which can make the GNSS-VIO system run in real-time and improve the positioning accuracy.We demonstrate that the states represented in the ECEF frame are fully observable, and the tightly coupled GNSS-VIO state estimator is consistent.

The rest of this paper is organized as follows: Section 2 introduces the implementation method of GNSS-VIO in detail, including line features, GNSS raw measurements processing, and observability analysis. Section 3 summarizes the detailed structure of our system. Section 4 conducts experiments on public datasets. Finally, conclusions are given in Section 5.

## 2. Methods

### 2.1. Frames and Notations

The frames involved in our system consist of:Sensor Frame: In our system, ${\left(\cdot \right)}^{c}$, ${\left(\cdot \right)}^{b}$, and ${\left(\cdot \right)}^{r}$ denote the camera frame, the body frame, and the GNSS receiver frame respectively.Local World Frame: The origin of the local world frame ${\left(\cdot \right)}^{w}$ is the position where the VIO system starts running, and the *z*-axis is gravity aligned as illustrated in Figure 1.ENU Frame: The ENU frame is also called the East–North–Up frame. The *x-*, *y-*, and *z*-axis of the ENU frame point to the east, north, and up directions, respectively. In our system, the ENU frame is at the same origin as the local world, and the *z*-axis of the two frames is aligned (Figure 1). We use ${\left(\cdot \right)}^{N}$ to represent the ENU frame.Geodetic Frame: As shown in Figure 1, geodetic coordinate ${\left(\cdot \right)}^{G}$ of a point **p** is represented by geodetic longitude *L*, geodetic latitude *B*, and geodetic height *H*. The geodetic longitude *L* of **p** is the angle between the reference meridian and another meridian that passes through **p**. The geodetic latitude *B* of **p** is the angle between the ellipsoid normal vector that passes through **p** and the projection of the ellipsoid normal vector into the equatorial plane. The geodetic height *H* of **p** is the minimum distance between **p** and the reference ellipsoid.ECEF Frame: The Earth-Centered, Earth-Fixed (ECEF) frame ${\left(\cdot \right)}^{E}$ is fixed to Earth. As depicted in Figure 1, the origin of the ECEF frame is at the center of mass of Earth. The *x*-axis points to the intersection of the Equator and the Prime Meridian. The *z*-axis is perpendicular to the equatorial plane in the direction of the North Pole. The *y*-axis is chosen to form a right-handed coordinate system with the *x-* and *z*-axis.


In this paper, $R_{b}^{w}$ and $p_{b}^{w}$ represent the rotation and translation of the body frame with respect to the local world frame, and $q_{b}^{w}$ is the corresponding quaternion form of the rotation $R_{b}^{w}$. $v_{b}^{w}$, $b_{a}$, and $b_{g}$ denote the velocity of the origin of the body frame measured in the local world frame, accelerometer bias, and gyroscope bias, respectively. $p_{c}^{b}$ and $q_{c}^{b}$ stand for the extrinsic parameters between the camera and IMU. $\delta t_{r}$ and $\delta {\dot{t}}_{r}$ represent the receiver clock error and receiver clock drifting rate, respectively. φ is the yaw between the local world frame and the ENU frame. $R_{w}^{N}$ is the rotation matrix of the local world frame with respect to the ENU frame. $p_{N}^{E}$ denotes the translation of the ENU frame with respect to the ECEF frame. $p_{r}^{b}$ is the extrinsic translation parameter between the GNSS receiver and IMU. ${\left[\cdot \right]}_{\times }$ represents the skew-symmetric matrix of a 3D vector.

### 2.2. Line Feature

#### 2.2.1. Plücker Coordinates

In our system, we described a spatial line with the Plücker Coordinates. Given a spatial line Lw∈πw, its Plücker Coordinates are represented by Lw=(nwT,dwT)T, where nw∈ℝ3 is the normal vector of the plane determined by Lw, and the origin of πw and dw∈ℝ3 is the line direction vector. We can transform Lw in the local world frame to Lc in the camera frame by:(1)Lc=Rwc[pwc]×Rwc0RwcLw=TwcLw,

#### 2.2.2. Line Feature Triangulation

Assuming that $L_{w}$ is observed by camera $c_{i}$ and camera $c_{j}$ in the normalized image plane as $z_{L_{w}}^{c_{i}}$ and $z_{L_{w}}^{c_{j}}$, the two line segments can be denoted by two endpoints as shown in Figure 2. The two endpoints of $z_{L_{w}}^{c_{i}}$ are $s_{L_{w}}^{c_{i}}={\left[u_{s}^{c_{i}},v_{s}^{c_{i}},1\right]}^{T}$ and $e_{L_{w}}^{c_{i}}={\left[u_{e}^{c_{i}},v_{e}^{c_{i}},1\right]}^{T}$, and the two endpoints of $z_{L_{w}}^{c_{j}}$ are $s_{L_{w}}^{c_{j}}={\left[u_{s}^{c_{j}},v_{s}^{c_{j}},1\right]}^{T}$ and $e_{L_{w}}^{c_{j}}={\left[u_{e}^{c_{j}},v_{e}^{c_{j}},1\right]}^{T}$. 

A 3D plane π can be modeled as:(2)$$\pi_{x}x+\pi_{y}y+\pi_{z}z+\pi_{w}=0,$$
where $n_{\pi }={\left[\pi_{x},\pi_{y},\pi_{z}\right]}^{T}$ is the normal vector of π. For a point $p_{0}={\left[x_{0},y_{0},z_{0}\right]}^{T}$ on the plane, we can obtain:(3)$$\pi_{w}=-\left(\pi_{x}x_{0}+\pi_{y}y_{0}+\pi_{z}z_{0}\right)=-n_{\pi }^{T}\cdot p_{0},$$

According to Equations (2) and (3), we can obtain the plane $\pi_{i}=\left(n_{\pi_{i}},\pi_{w_{i}}\right)$ and $\pi_{j}=\left(n_{\pi_{j}},\pi_{w_{j}}\right)$. As shown in Figure 2, the normal vector of $\pi_{i}$ is $n_{\pi_{i}}={\left[s_{L_{w}}^{c_{i}}\right]}_{\times }e_{L_{w}}^{c_{i}}$, and $\pi_{w_{i}}$ can be computed by the optical center $c_{i}={\left[0,0,0\right]}^{T}$ as $\pi_{w_{i}}=-n_{\pi_{i}}^{T}\cdot c_{0}=0$. To obtain $\pi_{j}$, we need to transform the two endpoints $s_{L_{w}}^{c_{j}}$ and $e_{L_{w}}^{c_{j}}$ to the camera frame $c_{i}$. Therefore, the corresponding reprojected endpoints are ${\tilde{s}}_{L_{w}}^{c_{j}}=R_{c_{j}}^{c_{i}}s_{L_{w}}^{c_{j}}+p_{c_{j}}^{c_{i}}$ and ${\tilde{e}}_{L_{w}}^{c_{j}}=R_{c_{j}}^{c_{i}}e_{L_{w}}^{c_{j}}+p_{c_{j}}^{c_{i}}$, where $R_{c_{j}}^{c_{i}}$ and $p_{c_{j}}^{c_{i}}$ can be obtained by visual-inertial alignment. Similarly, the normal vector of $\pi_{j}$ is $n_{\pi_{j}}={\left[{\tilde{s}}_{L_{w}}^{c_{j}}\right]}_{\times }{\tilde{e}}_{L_{w}}^{c_{j}}$, and $\pi_{w_{j}}$ can be calculated by the translation $p_{c_{j}}^{c_{i}}$ from the camera frame $c_{j}$ to the camera frame $c_{i}$ as $\pi_{w_{j}}=-n_{\pi_{j}}^{T}\cdot p_{c_{j}}^{c_{i}}$.

After $\pi_{i}$ and $\pi_{j}$ are computed, we can obtain the Plücker Coordinates of $L_{w}$ according to the dual Plücker matrix $L_{w}^{*}$:(4)$$L_{w}^{*}=\left[\begin{array}{cc} {\left[d_{w}\right]}_{\times } & n_{w}\\ -n_{w}^{T} & 0 \end{array}\right]=\pi_{i}\pi_{j}^{T}-\pi_{j}\pi_{i}^{T},$$

#### 2.2.3. Orthonormal Representation

Since spatial lines only have 4-DoF, there is a problem of overparameterization using the Plücker Coordinates to represent spatial lines. In contrast, the orthonormal representation (U,W)∈sO(3)×sO(2) is more suitable for nonlinear optimization. Let U=R(ψ) and W=R(ϕ) denote 3D and 2D rotation matrix, respectively, then we have:(5)$$U=R(\psi )=\left[u_{1},u_{2},u_{3}\right]=\left[\frac{n_{w}}{\left\|n_{w}\right\|},\frac{d_{w}}{\left\|d_{w}\right\|},\frac{n_{w}\times d_{w}}{\left\|n_{w}\times d_{w}\right\|}\right],$$
(6)$$\begin{array}{l} W=R(\phi )=\left[\begin{array}{cc} w_{1} & -w_{2}\\ w_{2} & w_{1} \end{array}\right]=\left[\begin{array}{cc} \cos(\phi ) & -\sin(\phi )\\ \sin(\phi ) & \cos(\phi ) \end{array}\right]\\ =\frac{1}{\sqrt{\left({\left\|n_{w}\right\|}^{2}+{\left\|d_{w}\right\|}^{2}\right)}}\left[\begin{array}{cc} \left\|n_{w}\right\| & -\left\|d_{w}\right\|\\ \left\|d_{w}\right\| & \left\|n_{w}\right\| \end{array}\right], \end{array}$$
where $\psi ={\left[\psi_{1},\psi_{2},\psi_{3}\right]}^{T}$ and ϕ are the 3D rotation angles around the *x-*, *y-*, and *z*-axis of the camera frame and the 2D rotation angle.

Therefore, we can define the orthonormal representation of a spatial line by a four-dimensional vector:(7)$$o={\left(\psi ,\phi \right)}^{T},$$

In addition, given an orthonormal representation (U,W), the corresponding Plücker Coordinates can be obtained by:(8)$${L^{\prime }}_{w}={\left[w_{1}u_{1}^{T},w_{2}u_{2}^{T}\right]}^{T}=\frac{1}{\sqrt{{\left\|n\right\|}^{2}+{\left\|d\right\|}^{2}}}L_{w},$$
where $w_{1}=\cos(\phi )$, $w_{2}=\sin(\phi )$, and $u_{i}$ is the *i*th column of U. $L_{w}$ and ${L^{\prime }}_{w}$ have a scale factor, but they denote the same spatial line.

#### 2.2.4. Line Feature Reprojection Residual

The line feature reprojection residual is defined in terms of point-to-line distance. Given a spatial line $L_{c}={\left(n_{c}^{T},d_{c}^{T}\right)}^{T}$, it can be projected to the image plane by [24]:(9)$$l={\left[l_{1},l_{2},l_{3}\right]}^{T}=K_{L}n_{c},$$
where $K_{L}$ is the line projection matrix.

Finally, assume that $L_{w}^{j}$ represents the *j*th spatial line $L_{j}$, which is observed by the *i*th camera frame $c_{i}$. Then the spatial line reprojection residual can be modeled as:(10)$$r_{L}\left({\tilde{z}}_{L_{j}}^{c_{i}},\chi \right)=\left[\begin{array}{c} d\left(s_{L_{j}}^{c_{i}},l_{L_{j}}^{c_{i}}\right)\\ d\left(e_{L_{j}}^{c_{i}},l_{L_{j}}^{c_{i}}\right) \end{array}\right]$$
where $s_{L_{j}}^{c_{i}}={\left[u_{s_{j}}^{c_{i}},v_{s_{j}}^{c_{i}},1\right]}^{T}$ and $e_{L_{j}}^{c_{i}}={\left[u_{e_{j}}^{c_{i}},v_{e_{j}}^{c_{i}},1\right]}^{T}$ denote the two endpoints of the line segment projected on the normalized image plane. d(s,l) is the point-to-line distance:(11)$$d(s,l)=\frac{s^{T}l}{\sqrt{l_{1}^{2}+l_{2}^{2}}},$$

For the corresponding Jacobian matrix, we followed the routine of [10].

### 2.3. GNSS Measurements

GNSS measurements include pseudorange, carrier phase, and Doppler frequency shift.

#### 2.3.1. Pseudorange Measurement

The pseudorange is defined as the measured distance obtained by multiplying the travel time of the satellite signal by the speed of light. Due to the influence of satellite clock error, receiver clock error, and ionospheric and tropospheric delays, the pseudorange is prefixed with “pseudo” to distinguish it from the true distance from the satellite to the GNSS receiver. Generally, although the pseudorange measurement only has meter-level positioning precision (the positioning precision of P code is about 10 m, and the positioning precision of C/A code is 20 m to 30 m), it is real-time and has no ambiguity. Therefore, the pseudorange measurement is still very important for GNSS positioning technology. For a certain satellite $s_{j}$ and a GNSS receiver $r_{k}$ at time *k*, the pseudorange ${\tilde{P}}_{r_{k}}^{s_{j}}$ can be modeled as:(12)$${\tilde{P}}_{r_{k}}^{s_{j}}=\left\|p_{s_{j}}^{E}-p_{r_{k}}^{E}\right\|+c\left(\delta t_{r_{k}}-\delta t_{s_{j}}\right)+T_{r_{k}}^{s_{j}}+I_{r_{k}}^{s_{j}}+\varepsilon_{p_{k}^{j}},$$
where $p_{s_{j}}^{E}$ and $p_{r_{k}}^{E}$ represent the position of satellite $s_{j}$ and the GNSS receiver $r_{k}$ at time *k* in the ECEF frame, respectively. c is the speed of light. $\delta t_{s_{j}}$, $T_{r_{k}}^{s_{j}}$, and $I_{r_{k}}^{s_{j}}$ are the satellite clock error and tropospheric and ionospheric delays, which can be computed according to the satellite ephemeris. $\varepsilon_{p_{k}^{j}}∼\left(0,\sigma_{p_{k}^{j}}^{2}\right)$ denotes the multipath and random error of the pseudorange measurement, which is subject to the Zero-mean Gaussian distribution.

#### 2.3.2. Carrier Phase Measurement

Although pseudorange positioning is an important method for GNSS, its error is too large for some applications that require high-precision positioning. In contrast, due to the short wavelength of the carrier phase ($\lambda_{L_{1}}=19\mathrm{cm}$ and $\lambda_{L_{2}}=24\mathrm{cm}$), if the carrier phase measurement is used for positioning, it can achieve centimeter-level positioning accuracy. However, since the carrier phase is a periodic sinusoidal signal, and the GNSS receiver can only measure the part of less than one wavelength, there is the problem of the whole-cycle ambiguity, which makes the positioning process time-consuming. The carrier phase is defined as the phase difference between the phase transmitted by the satellite and the reference phase generated by the GNSS receiver. Similar to the pseudorange measurement, the carrier phase measurement is also related to the position of satellite and GNSS receiver. The carrier phase measurement can be modeled as:(13)$${\tilde{\Phi }}_{r_{k}}^{s_{j}}=\left\|p_{s_{j}}^{E}-p_{r_{k}}^{E}\right\|+c\left(\delta t_{r_{k}}-\delta t_{s_{j}}\right)+T_{r_{k}}^{s_{j}}-I_{r_{k}}^{s_{j}}+\lambda N+\varepsilon_{\Phi_{k}^{j}},$$
where λ is the carrier wavelength, and N is the whole-cycle ambiguity. $\varepsilon_{\Phi_{k}^{j}}∼\left(0,\sigma_{\Phi_{k}^{j}}^{2}\right)$ represents the multipath and random error of the carrier phase measurement, which is subject to the Zero-mean Gaussian distribution.

#### 2.3.3. Doppler Frequency Shift Measurement

The Doppler effect reveals a phenomenon in the spread of waves, that is, the wavelength of the radiation emitted by objects changes accordingly due to the relative motion of the wave source and the observer. As the wave source moves toward the observer, the wavelength becomes shorter and the frequency becomes higher due to the compression of the wave. On the contrary, as the wave source moves away from the observer, the wavelength becomes longer and the frequency becomes lower. Similarly, the Doppler effect can also occur between the satellite and the GNSS receiver. When a satellite orbits Earth in its elliptical orbit, due to the relative motion between the satellite and the GNSS receiver, the frequency of the satellite signal received by the GNSS receiver changes accordingly. This frequency change is called Doppler frequency shift, which can be modeled as:(14)$$\Delta {\tilde{f}}_{r_{k}}^{s_{j}}=-\frac{1}{\lambda }\left[\kappa_{k}^{T}\left(v_{s_{j}}^{E}-v_{r_{k}}^{E}\right)+c\left(\delta {\dot{t}}_{r_{k}}-\delta {\dot{t}}_{s_{j}}\right)\right]+\varepsilon_{\Delta f_{k}^{j}},$$
where $\kappa_{k}=\frac{\left(p_{s_{j}}^{E}-p_{r_{k}}^{E}\right)}{\left\|p_{s_{j}}^{E}-p_{r_{k}}^{E}\right\|}$ is the unit vector pointing along the line of sight from the GNSS receiver $r_{k}$ to the satellite $s_{j}$. $v_{s_{j}}^{E}$ and $v_{r_{k}}^{E}$ denote the velocity of the *j*th satellite and the receiver at time *k* in the ECEF frame, respectively. $\delta {\dot{t}}_{s_{j}}$ is the satellite clock drifting rate, which can be calculated according to the satellite ephemeris. $\varepsilon_{\Delta f_{k}^{j}}∼\left(0,\sigma_{\Delta f_{k}^{j}}^{2}\right)$ represents the multipath and random error of the Doppler frequency shift measurement, which is subject to the Zero-mean Gaussian distribution.

### 2.4. Carrier Phase Smoothed Pseudorange

As mentioned above, the pseudorange measurement proves fast and efficient, but can only achieve meter-level positioning precision. In contrast, the carrier phase measurement can achieve centimeter-level positioning precision but is required to solve the whole-cycle ambiguity, which is complex and time-consuming. Therefore, to combine the merits of the two measurements, we can use carrier phase smoothed pseudorange (short for smoothed pseudorange) to improve the positioning precision. In general, the positioning precision of the smoothed pseudorange measurement is several times higher than that of the pseudorange measurement. For a single-frequency receiver, when the whole-cycle ambiguity and ionospheric delay are nearly constant within a period of time, the pseudorange can be smoothed by using the carrier phase. The Hatch filter is the most widely used for carrier phase smoothed pseudorange, which assumes that the ionospheric delay is nearly constant among GNSS epochs and then averages the multiepoch whole-cycle ambiguity and ionospheric delay to improve the positioning accuracy of the pseudorange measurement. The carrier phase smoothed pseudorange based on the Hatch filter can be modeled as:(15)$${\bar{\rho }}_{r_{k}}^{s_{j}}=\frac{1}{k_{m}}{\tilde{P}}_{r_{k}}^{s_{j}}+\left(1-\frac{1}{k_{m}}\right)\left[{\bar{\rho }}_{r_{k-1}}^{s_{j}}+{\tilde{\Phi }}_{r_{k}}^{s_{j}}-{\tilde{\Phi }}_{r_{k-1}}^{s_{j}}\right]+\varepsilon_{\rho_{k}^{j}},1\leq k_{m}\leq m,$$
where *m* is the smoothed interval of the Hatch filter, usually 20 to 200 epochs. $\varepsilon_{\rho_{k}^{j}}∼\left(0,\sigma_{\rho_{k}^{j}}^{2}\right)$ denotes the multipath and random error of the carrier phase smoothed pseudorange measurement, which is also subject to the Zero-mean Gaussian distribution. According to Equation (15), the variance of the smoothed pseudorange is related to the variance of the pseudorange and carrier phase measurements. Assuming that the variance at different times is independent, then the variance of the smoothed pseudorange can be modeled as:(16)$$\sigma_{\rho_{k}^{j}}^{2}=\frac{1}{k_{m}}\sigma_{P_{k}^{j}}^{2}+\left(1-\frac{1}{k_{m}}\right)\left(\sigma_{\Phi_{k}^{j}}^{2}+\sigma_{\Phi_{k-1}^{j}}^{2}\right),$$

### 2.5. Factor Graph Representation

The factor graph of our system is shown in Figure 3, which shows the factors in a sliding window, including point feature factors, line feature factors, IMU factors, carrier phase smoothed pseudorange factors, and Doppler frequency shift factors. Visual observation consists of the point and line features detected by our system. In nonlinear optimization, the states are optimized according to the residuals of these factors. The states of our system include:(17)$$\begin{array}{l} X=\left[x_{0},x_{1},\cdots x_{n},\mu_{0},\mu_{1},\cdots \mu_{i},o_{0},o_{1},\cdots ,o_{j},p_{c}^{b},q_{c}^{b},\varphi ,p_{N}^{E},p_{r}^{b}\right],\\ x_{k}=\left[p_{b_{k}}^{w},v_{b_{k}}^{w},q_{b_{k}}^{w},b_{a},b_{g},\delta t_{r_{k}},\delta {\dot{t}}_{r_{k}}\right],k=0,1,\cdots ,n,\\ o=\left[\psi ,\phi \right], \end{array}$$
where *n* is the sliding window size, *i* is the number of point features, and *j* is the number of line features. $\mu_{i}$ is the inverse depth of the *i*th point feature in the sliding window.

The IMU preintegration and point feature residuals can be obtained according to [16], and the line feature residual can be obtained from Equation (10). In the following, we compute the smoothed pseudorange and Doppler frequency shift residuals in detail. The position of the receiver at time *k* in the ECEF frame is:(18)$$p_{r_{k}}^{E}=R_{N}^{E}R_{w}^{N}p_{r_{k}}^{w}+p_{N}^{E},$$
where $p_{r_{k}}^{w}$ is the position of the receiver in the local world frame, and it can be modeled as:(19)$$p_{r_{k}}^{w}=R_{b_{k}}^{w}p_{r}^{b}+p_{b_{k}}^{w},$$

$R_{w}^{N}$ is the rotation matrix of the local world frame with respect to the ENU frame. Since the two frames are gravity-aligned, the 1-DoF $R_{w}^{N}$ is only related to the yaw offset ϕ and can be modeled as:(20)$$R_{w}^{N}=\left[\begin{array}{ccc} cos\phi & -sin\phi & 0\\ sin\phi & cos\phi & 0\\ 0 & 0 & 1 \end{array}\right],$$

$R_{N}^{E}$ represents the rotation matrix of the ENU frame with respect to the ECEF frame, which is determined by the longitude *L* and latitude *B* of $p_{N}^{E}$. Given a position $p_{N}^{E}={\left[X,Y,Z\right]}^{T}$ in the ECEF frame, it can be denoted in the geodetic frame as:(21)$$\begin{array}{l} \left\{\begin{array}{l} L=\arctan\left(Y/X\right)\\ B=\arctan\left\{Z\left(N_{r}+H\right)/\left[\sqrt{X^{2}+Y^{2}}\left(N_{r}\left(1-e^{2}\right)+H\right)\right]\right\}\\ H=Z/\sin B-N_{r}\left(1-e^{2}\right) \end{array}\right.,\\ N_{r}=a/\sqrt{1-e^{2}\sin^{2}B},e^{2}=\left(a^{2}-b^{2}\right)/a^{2}, \end{array}$$
where *a* and *b* denote the semimajor axis and the semiminor axis of the elliptical orbit, respectively. $N_{r}$ is the radius of curvature in prime vertical, and *e* is the eccentricity that is related to *a* and *b*.

After the longitude *L* and latitude *B* of $p_{N}^{E}$ are computed, $R_{N}^{E}$ can be obtained by:(22)$$R_{N}^{E}=\left[\begin{array}{ccc} -\sin L & -\sin B\cos L & \cos B\cos L\\ \cos L & -\sin B\sin L & \cos B\sin L\\ 0 & \cos B & \sin B \end{array}\right],$$

According to Equation (15), the smoothed pseudorange residual can be represented as:(23)$$r_{\rho }({\tilde{z}}_{r_{k}}^{s_{j}},X)=\frac{1}{k_{m}}{\tilde{P}}_{r_{k}}^{s_{j}}+\left(1-\frac{1}{k_{m}}\right)\left[{\bar{\rho }}_{r_{k-1}}^{s_{j}}+{\tilde{\Phi }}_{r_{k}}^{s_{j}}-{\tilde{\Phi }}_{r_{k-1}}^{s_{j}}\right]-{\bar{\rho }}_{r_{k}}^{s_{j}},$$

In nonlinear optimization, it is necessary to calculate the Jacobian matrix of the smoothed pseudorange residual with respect to the states. Therefore, from Equation (23), the Jacobian matrix of the smoothed pseudorange can be obtained as:(24)$$J_{\rho }=\left[\begin{array}{ccccccccc} \frac{\partial r_{\rho }}{\partial p_{b_{k-1}}^{w}} & \frac{\partial r_{\rho }}{\partial R_{b_{k-1}}^{w}} & \frac{\partial r_{\rho }}{\partial p_{b_{k}}^{w}} & \frac{\partial r_{\rho }}{\partial R_{b_{k}}^{w}} & \frac{\partial r_{\rho }}{\partial \delta t_{r_{k-1}}} & \frac{\partial r_{\rho }}{\partial \delta t_{r_{k}}} & \frac{\partial r_{\rho }}{\partial \varphi } & \frac{\partial r_{\rho }}{\partial p_{N}^{E}} & \frac{\partial r_{\rho }}{\partial p_{r}^{b}} \end{array}\right],$$
with
(25)$$\begin{array}{l} \frac{\partial r_{\rho }}{\partial p_{b_{k-1}}^{w}}=\left(1-\frac{1}{k_{m}}\right)\frac{{\left(p_{s_{j}}^{E}-p_{r_{k-1}}^{E}\right)}^{T}}{\left\|p_{s_{j}}^{E}-p_{r_{k-1}}^{E}\right\|}R_{N}^{E}R_{w}^{N},\\ \frac{\partial r_{\rho }}{\partial R_{b_{k-1}}^{w}}=-\left(1-\frac{1}{k_{m}}\right)\frac{{\left(p_{s_{j}}^{E}-p_{r_{k-1}}^{E}\right)}^{T}}{\left\|p_{s_{j}}^{E}-p_{r_{k-1}}^{E}\right\|}R_{N}^{E}R_{w}^{N}R_{b_{k-1}}^{w}{\left[p_{r}^{b}\right]}_{\times },\\ \frac{\partial r_{\rho }}{\partial p_{b_{k}}^{w}}=-\frac{{\left(p_{s_{j}}^{E}-p_{r_{k}}^{E}\right)}^{T}}{\left\|p_{s_{j}}^{E}-p_{r_{k}}^{E}\right\|}R_{N}^{E}R_{w}^{N},\\ \frac{\partial r_{\rho }}{\partial R_{b_{k}}^{w}}=\frac{{\left(p_{s_{j}}^{E}-p_{r_{k}}^{E}\right)}^{T}}{\left\|p_{s_{j}}^{E}-p_{r_{k}}^{E}\right\|}R_{N}^{E}R_{w}^{N}R_{b_{k}}^{w}{\left[p_{r}^{b}\right]}_{\times },\\ \frac{\partial r_{\rho }}{\partial \delta t_{r_{k-1}}}=-c\left(1-\frac{1}{k_{m}}\right),\\ \frac{\partial r_{\rho }}{\partial \delta t_{r_{k}}}=c,\\ \frac{\partial r_{\rho }}{\partial \varphi }=-\frac{{\left(p_{s_{j}}^{E}-p_{r_{k}}^{E}\right)}^{T}}{\left\|p_{s_{j}}^{E}-p_{r_{k}}^{E}\right\|}R_{N}^{E}{\dot{R}}_{w}^{N}p_{r_{k}}^{w}+\left(1-\frac{1}{k_{m}}\right)\frac{{\left(p_{s_{j}}^{E}-p_{r_{k-1}}^{E}\right)}^{T}}{\left\|p_{s_{j}}^{E}-p_{r_{k-1}}^{E}\right\|}R_{N}^{E}{\dot{R}}_{w}^{N}p_{r_{k-1}}^{w},\\ \frac{\partial r_{\rho }}{\partial p_{N}^{E}}=-\frac{{\left(p_{s_{j}}^{E}-p_{r_{k}}^{E}\right)}^{T}}{\left\|p_{s_{j}}^{E}-p_{r_{k}}^{E}\right\|}+\left(1-\frac{1}{k_{m}}\right)\frac{{\left(p_{s_{j}}^{E}-p_{r_{k-1}}^{E}\right)}^{T}}{\left\|p_{s_{j}}^{E}-p_{r_{k-1}}^{E}\right\|},\\ \frac{\partial r_{\rho }}{\partial p_{r}^{b}}=-\frac{{\left(p_{s_{j}}^{E}-p_{r_{k}}^{E}\right)}^{T}}{\left\|p_{s_{j}}^{E}-p_{r_{k}}^{E}\right\|}R_{N}^{E}R_{w}^{N}R_{b_{k}}^{w}+\left(1-\frac{1}{k_{m}}\right)\frac{{\left(p_{s_{j}}^{E}-p_{r_{k-1}}^{E}\right)}^{T}}{\left\|p_{s_{j}}^{E}-p_{r_{k-1}}^{E}\right\|}R_{N}^{E}R_{w}^{N}R_{b_{k-1}}^{w}, \end{array}$$
where
(26)$${\dot{R}}_{w}^{N}=\frac{dR_{w}^{N}}{d\varphi }=\left[\begin{array}{ccc} -sin\phi & -cos\phi & 0\\ cos\phi & -sin\phi & 0\\ 0 & 0 & 0 \end{array}\right],$$

The derivation details are provided in Appendix A.

In Equation (14), the velocity of the receiver in the ECEF frame is transformed from that in the local world frame:(27)$$v_{r_{k}}^{E}≃R_{N}^{E}R_{w}^{N}v_{b_{k}}^{w},$$

Therefore, the Doppler frequency shift residual can be obtained by:(28)$$r_{\Delta f}({\tilde{z}}_{r_{k}}^{s_{j}},X)=\frac{1}{\lambda }\left[\kappa_{k}^{T}\left(v_{s_{j}}^{E}-v_{r_{k}}^{E}\right)+c\left(\delta {\dot{t}}_{r_{k}}-\delta {\dot{t}}_{s_{j}}\right)\right]+\Delta {\tilde{f}}_{r_{k}}^{s_{j}},$$

Similar to the smoothed pseudorange, the Jacobian matrix of the Doppler frequency shift is:(29)$$J_{\Delta f}=\left[\begin{array}{ccccccc} \frac{\partial r_{\Delta f}}{\partial p_{b_{k}}^{w}} & \frac{\partial r_{\Delta f}}{\partial R_{b_{k}}^{w}} & \frac{\partial r_{\Delta f}}{\partial v_{b_{k}}^{w}} & \frac{\partial r_{\Delta f}}{\partial \delta {\dot{t}}_{r_{k}}} & \frac{\partial r_{\Delta f}}{\partial \varphi } & \frac{\partial r_{\Delta f}}{\partial p_{N}^{E}} & \frac{\partial r_{\Delta f}}{\partial p_{r}^{b}} \end{array}\right]$$
with


(30)
$$\begin{array}{l} \frac{\partial r_{\Delta f}}{\partial p_{b_{k}}^{w}}=-{\left(v_{s_{j}}^{E}-v_{r_{k}}^{E}\right)}^{T}\frac{I{\left\|p_{s_{j}}^{E}-p_{r_{k}}^{E}\right\|}^{2}-\left(p_{s_{j}}^{E}-p_{r_{k}}^{E}\right){\left(p_{s_{j}}^{E}-p_{r_{k}}^{E}\right)}^{T}}{\lambda {\left\|p_{s_{j}}^{E}-p_{r_{k}}^{E}\right\|}^{3}}R_{N}^{E}R_{w}^{N},\\ \frac{\partial r_{\Delta f}}{\partial R_{b_{k}}^{w}}={\left(v_{s_{j}}^{E}-v_{r_{k}}^{E}\right)}^{T}\frac{I{\left\|p_{s_{j}}^{E}-p_{r_{k}}^{E}\right\|}^{2}-\left(p_{s_{j}}^{E}-p_{r_{k}}^{E}\right){\left(p_{s_{j}}^{E}-p_{r_{k}}^{E}\right)}^{T}}{\lambda {\left\|p_{s_{j}}^{E}-p_{r_{k}}^{E}\right\|}^{3}}R_{N}^{E}R_{w}^{N}R_{b_{k}}^{w}{\left[p_{r}^{b}\right]}_{\times },\\ \frac{\partial r_{\Delta f}}{\partial v_{b_{k}}^{w}}=-\frac{1}{\lambda }\kappa_{k}^{T}R_{N}^{E}R_{w}^{N},\\ \frac{\partial r_{\Delta f}}{\partial \delta {\dot{t}}_{r_{k}}}=\frac{c}{\lambda },\\ \frac{\partial r_{\Delta f}}{\partial \phi }\approx -\frac{1}{\lambda }\kappa_{k}^{T}R_{N}^{E}{\dot{R}}_{w}^{N}v_{b_{k}}^{w},\\ \frac{\partial r_{\Delta f}}{\partial p_{N}^{E}}\approx 0,\\ \frac{\partial r_{\Delta f}}{\partial p_{r}^{b}}=-{\left(v_{s_{j}}^{E}-v_{r_{k}}^{E}\right)}^{T}\frac{I{\left\|p_{s_{j}}^{E}-p_{r_{k}}^{E}\right\|}^{2}-\left(p_{s_{j}}^{E}-p_{r_{k}}^{E}\right){\left(p_{s_{j}}^{E}-p_{r_{k}}^{E}\right)}^{T}}{\lambda {\left\|p_{s_{j}}^{E}-p_{r_{k}}^{E}\right\|}^{3}}R_{N}^{E}R_{w}^{N}R_{b_{k}}^{w}, \end{array}$$


The corresponding derivation rule is similar to the Jacobian matrix of the smoothed pseudorange and will not be repeated here.

After the residuals of all factors are obtained, then the system can optimize the states by Ceres solver [25]. The cost function of our system is:(31)$$\begin{array}{c} \underset{X}{\min}\left\{{\left\|r_{p}-H_{p}X\right\|}^{2}+\sum_{k\in B}{\left\|r_{B}({\tilde{z}}_{b_{k+1}}^{b_{k}},X)\right\|}_{P_{b_{k+1}}^{b_{k}}}^{2}+\sum_{(l,j)\in C}\alpha \left({\left\|r_{C}({\tilde{z}}_{l}^{c_{j}},X)\right\|}_{P_{l}^{c_{j}}}^{2}\right)\right.\\ +\underset{\left(i,j\right)\in L}{\sum }\alpha \left(r_{L}{\left(z{˜}_{{}_{j}}^{c_{i}},\right)}_{P_{{}_{j}}^{c_{i}}}^{2}\right)+\sum_{(k,j)\in \rho }{\left\|r_{\rho }({\tilde{z}}_{r_{k}}^{s_{j}},X)\right\|}_{\sigma_{\rho }^{2}}^{2}\left.+\sum_{(k,j)\in \Delta f}{\left\|r_{\Delta f}({\tilde{z}}_{r_{k}}^{s_{j}},X)\right\|}_{\sigma_{\Delta f}^{2}}^{2}\right\}, \end{array}$$
where $r_{p}$ is the prior residual for marginalization. B is the set of IMU preintegration measurements in the sliding window. C and L are the set of point and line features in the sliding window, respectively. α is the Cauchy robust function used to suppress outliers.

### 2.6. GNSS-IMU Calibration

$p_{r}^{b}$ is the extrinsic translation parameter between the GNSS receiver and IMU. After our system performs ENU Origin Estimation and Yaw Estimation, then we can calibrate $p_{r}^{b}$. We estimate $p_{r}^{b}$ as the initial value of nonlinear optimization through the following optimization problem:(32)$$\underset{p_{r}^{b}}{\min}\sum_{(k,j)\in \Delta f}{\left\|r_{\Delta f}({\tilde{z}}_{r_{k}}^{s_{j}},X)\right\|}_{\sigma_{\Delta f}^{2}}^{2},$$

After the GNSS-IMU calibration is successfully initialized, our system performs the nonlinear optimization.

### 2.7. Observability Analysis of Tightly Coupled GNSS-VIO System

A SLAM system can be described using a state equation and an output equation, where the input and output are the external variables of the system, and the state is the internal variables of the system. If the states of the system can be completely represented by the output, the system is fully observable; otherwise, the system is not fully observable. Observability plays a very important role in the state estimation problem. If some states of the system are not observable, the positioning precision of the system will be affected when running in long trajectories. The observability of the system can be represented by the observability matrix. If the dimension of the null space of the observability matrix is equal to 0, then the system is fully observable. To facilitate the observability analysis of our proposed system, some simplifications were required. First, the accelerometer and gyroscope biases were not included in the states, because the biases were observable and they did not change the results of the observability analysis [26]. Second, we considered a single point and line feature [27]. Third, the translation parameter $p_{r}^{b}$ was successfully calibrated. Then the discrete-time linear error state model and residual of the system are:(33)$$\begin{array}{l} \delta x_{k+1}≃\Phi_{k}\delta x_{k}+w_{k},\\ r_{k}=H_{k}\delta x_{k}+n_{k}, \end{array}$$
where $\delta x_{k}$ is the error state, and $r_{k}$ is the residual. $\Phi_{k}$ and $H_{k}$ are the error-state transition matrix and the measurement Jacobian matrix, respectively. $w_{k}$ and $n_{k}$ represent the system noise process and the measurement noise process, respectively. The noise process is modeled as a Zero-mean white Gaussian process.

According to [28], the observability matrix can be obtained as:(34)$$M=\left[\begin{array}{c} H_{k}\\ H_{k+1}\Phi_{k}\\ \vdots \\ H_{k+t}\Phi_{k+t-1}\cdots \Phi_{k} \end{array}\right],$$

In Equation (34), the observability matrix is defined as a function of the error-state transition matrix $\Phi_{k}$ and the measurement Jacobian matrix $H_{k}$.

Therefore, given the linearized system in Equation (33), its observability can be demonstrated according to Equation (34). The proof is as follows:

**Theorem** **1.***The states represented in the ECEF frame are fully observable*.

**Proof** **of** **Theorem****1.**The simplified states include:(35)$$x_{k}^{E}=\left[p_{b_{k}}^{E},v_{b_{k}}^{E},q_{b_{k}}^{E},\mu ,o,\phi ,p_{N}^{E}\right],$$Generally, the raw accelerometer and gyroscope measurements from IMU can be obtained by:(36)$$\begin{array}{l} {\tilde{a}}_{b}=R_{w}^{b}(a_{w}+g_{w})+b_{a}+n_{a},\\ {\tilde{\omega }}_{b}=\omega_{b}+b_{g}+n_{g}, \end{array}$$
where ${\tilde{a}}_{b}$ and ${\tilde{\omega }}_{b}$ are the accelerometer and gyroscope measurements. The measurements are represented in the local world frame.Given two time instants, position, velocity, and orientation states can be propagated by the IMU measurements:(37)$$\begin{array}{l} p_{b_{k+1}}^{w}=p_{b_{k}}^{w}+v_{b_{k}}^{w}\Delta t_{k}+\int \int_{t\in (t_{k},t_{k+1})}\left[R_{b_{t}}^{w}\left({\tilde{a}}_{b_{t}}-b_{a}\right)-g_{w}\right]dt^{2},\\ v_{b_{k+1}}^{w}=v_{b_{k}}^{w}+\int_{t\in (t_{k},t_{k+1})}\left[R_{b_{t}}^{w}\left({\tilde{a}}_{b_{t}}-b_{a}\right)-g_{w}\right]dt,\\ q_{b_{k+1}}^{w}=\int_{t\in (t_{k},t_{k+1})}q_{b_{t}}^{w}⊗\left[\begin{array}{c} 0\\ \frac{1}{2}\left({\tilde{\omega }}_{b_{t}}-b_{g}\right) \end{array}\right]dt, \end{array}$$
where ⊗ denotes the quaternion multiplication operation.Equation (37) is the continuous state propagation model. To analyze the observability of the system, it was necessary to perform the discretization of the continuous-time system model. Therefore, we used the Euler method to compute the discrete-time model of Equation (37):(38)$$\begin{array}{l} p_{b_{k+1}}^{w}=p_{b_{k}}^{w}+v_{b_{k}}^{w}\Delta t_{k}+\frac{1}{2}\left[R_{b_{k}}^{w}\left({\tilde{a}}_{b_{k}}-b_{a}\right)-g_{w}\right]\Delta t_{k}^{2},\\ v_{b_{k+1}}^{w}=v_{b_{k}}^{w}+\left[R_{b_{k}}^{w}\left({\tilde{a}}_{b_{k}}-b_{a}\right)-g_{w}\right]\Delta t_{k},\\ q_{b_{k+1}}^{w}=q_{b_{k}}^{w}⊗\left[\begin{array}{c} 1\\ \frac{1}{2}\left({\tilde{\omega }}_{b_{k}}-b_{g}\right)\Delta t_{k} \end{array}\right], \end{array}$$According to Equation (38), the error propagation equation of position, velocity, and orientation can be obtained by:(39)$$\left[\begin{array}{l} \delta p_{b_{k+1}}^{w}\\ \delta v_{b_{k+1}}^{w}\\ \delta \theta_{b_{k+1}}^{w} \end{array}\right]=\left[\begin{array}{lll} I & \Delta t_{k}I & -\frac{1}{2}R_{b_{k}}^{w}{\left[{\tilde{a}}_{b_{k}}-b_{a}\right]}_{\times }\Delta t_{k}^{2}\\ 0 & I & -R_{b_{k}}^{w}{\left[{\tilde{a}}_{b_{k}}-b_{a}\right]}_{\times }\Delta t_{k}\\ 0 & 0 & I-{\left[{\tilde{\omega }}_{b_{k}}-b_{g}\right]}_{\times }\Delta t_{k} \end{array}\right]\left[\begin{array}{c} \delta p_{b_{k}}^{w}\\ \delta v_{b_{k}}^{w}\\ \delta \theta_{b_{k}}^{w} \end{array}\right]+{w^{\prime }}_{k},$$The details of Equation (39) are provided in Appendix B.Similarly, when the states are represented in the ECEF frame, Equation (39) is still held, namely:(40)$$\left[\begin{array}{l} \delta p_{b_{k+1}}^{E}\\ \delta v_{b_{k+1}}^{E}\\ \delta \theta_{b_{k+1}}^{E} \end{array}\right]=\left[\begin{array}{lll} I & \Delta t_{k}I & -\frac{1}{2}R_{b_{k}}^{E}{\left[{\tilde{a}}_{b_{k}}-b_{a}\right]}_{\times }\Delta t_{k}^{2}\\ 0 & I & -R_{b_{k}}^{E}{\left[{\tilde{a}}_{b_{k}}-b_{a}\right]}_{\times }\Delta t_{k}\\ 0 & 0 & I-{\left[{\tilde{\omega }}_{b_{k}}-b_{g}\right]}_{\times }\Delta t_{k} \end{array}\right]\left[\begin{array}{c} \delta p_{b_{k}}^{E}\\ \delta v_{b_{k}}^{E}\\ \delta \theta_{b_{k}}^{E} \end{array}\right]+{w^{\prime }}_{k},$$Therefore, the discrete-time error state model of Equation (35) can be obtained:(41)$$\begin{array}{l} \left[\begin{array}{c} \delta p_{b_{k+1}}^{E}\\ \delta v_{b_{k+1}}^{E}\\ \delta \theta_{b_{k+1}}^{E}\\ \delta \mu_{k+1}\\ \delta o_{k+1}\\ \delta \varphi_{k+1}\\ \delta p_{N}^{E}{}_{\left(k+1\right)} \end{array}\right]=\left[\begin{array}{cccc} I_{3} & \Delta t_{k}I_{3} & \Phi_{k}^{1} & 0_{3\times 9}\\ 0_{3} & I_{3} & \Phi_{k}^{2} & 0_{3\times 9}\\ 0_{3} & 0_{3} & \Phi_{k}^{3} & 0_{3\times 9}\\ 0_{9\times 3} & 0_{9\times 3} & 0_{9\times 3} & I_{9} \end{array}\right]\left[\begin{array}{c} \delta p_{b_{k}}^{E}\\ \delta v_{b_{k}}^{E}\\ \delta \theta_{b_{k}}^{E}\\ \delta \mu_{k}\\ \delta o_{k}\\ \delta \varphi_{k}\\ \delta p_{N}^{E}{}_{\left(k\right)} \end{array}\right]+w_{k},\\ \Rightarrow \delta x_{k+1}^{E}=\Phi_{k}\delta x_{k}^{E}+w_{k}, \end{array}$$
where
(42)$$\left\{\begin{array}{l} \Phi_{k}^{1}=-\frac{1}{2}R_{b_{k}}^{E}{\left[{\tilde{a}}_{b_{k}}-b_{a}\right]}_{\times }\Delta t_{k}^{2}\\ \Phi_{k}^{2}=-R_{b_{k}}^{E}{\left[{\tilde{a}}_{b_{k}}-b_{a}\right]}_{\times }\Delta t_{k}\\ \Phi_{k}^{3}=I-{\left[{\tilde{\omega }}_{b_{k}}-b_{g}\right]}_{\times }\Delta t_{k} \end{array}\right.,$$Since $\Phi_{k}$ is an upper triangular matrix, the error-state transition matrix $\Phi_{k+t,k}$ from time step *k* to *k* + *t* is also an upper triangular matrix, namely:(43)$$\begin{array}{l} \Phi_{k+t,k}=\Phi_{k+t-1}\cdots \Phi_{k+1}\Phi_{k}\\ =\left[\begin{array}{llll} I_{3} & (t_{k+t-1}-t_{k-1})I_{3} & \Phi_{13}^{k+t} & 0_{3\times 9}\\ 0_{3} & I_{3} & \Phi_{23}^{k+t} & 0_{3\times 9}\\ 0_{3} & 0_{3} & \Phi_{33}^{k+t} & 0_{3\times 9}\\ 0_{9\times 3} & 0_{9\times 3} & 0_{9\times 3} & I_{9} \end{array}\right], \end{array}$$
where $\Phi_{13}^{k+t}$, $\Phi_{23}^{k+t}$, and $\Phi_{33}^{k+t}$ are nonzero entries.For the measurement Jacobian matrix, our system consists of visual observations and GNSS measurements. GNSS measurements include pseudorange, carrier phase, and Doppler frequency shift, and any of them yields the same results for observability analysis. Therefore, in the following, we used the smoothed pseudorange measurement for observability analysis.The Jacobian matrix of the smoothed pseudorange measurement with respect to the states in Equation (35) can be obtained by:(44)$$\begin{array}{ll} H_{k+t}^{g} & =\left[\begin{array}{llll} p_{k+t} & 0_{1\times 11} & p_{k+t}R_{N}^{E}{\dot{R}}_{w}^{N}p_{r_{k+t}}^{w}-\left(1-\frac{1}{k_{m}}\right)p_{k+t-1}R_{N}^{E}{\dot{R}}_{w}^{N}p_{r_{k+t-1}}^{w} & p_{k+t}-\left(1-\frac{1}{k_{m}}\right)p_{k+t-1} \end{array}\right]\\ & =\left[\begin{array}{cccc} h_{k+t}^{1} & 0_{1\times 11} & h_{k+t}^{2} & h_{k+t}^{3} \end{array}\right],\\ p_{k+t} & =-\frac{{\left(p_{s_{j}}^{E}-p_{r_{k+t}}^{E}\right)}^{T}}{\left\|p_{s_{j}}^{E}-p_{r_{k+t}}^{E}\right\|},p_{k+t-1}=-\frac{{\left(p_{s_{j}}^{E}-p_{r_{k+t-1}}^{E}\right)}^{T}}{\left\|p_{s_{j}}^{E}-p_{r_{k+t-1}}^{E}\right\|}, \end{array}$$The derivation details are similar to the Jacobian matrix of the smoothed pseudorange, which has been computed in Appendix A. Since the smoothed pseudorange measurement does not include velocity, orientation, the inverse depth of point feature, and the orthonormal representation of line feature, the corresponding entry of the Jacobian matrix is equal to zero.In addition, we can obtain the Jacobian matrix of the visual observations by:(45)$$H_{k+t}^{v}=\left[\begin{array}{cccc} 0_{2\times 9} & \frac{\partial r_{C}}{\partial \delta \mu_{k+t}} & 0_{2\times 4} & 0_{2\times 4}\\ 0_{2\times 9} & 0_{2\times 1} & \frac{\partial r_{L}}{\partial \delta o_{k+t}} & 0_{2\times 4} \end{array}\right]$$
where $\frac{\partial r_{C}}{\partial \delta \mu_{k+t}}$ and $\frac{\partial r_{L}}{\partial \delta o_{k+t}}$ can be obtained from [10].Therefore, we can obtain the entry of the observability matrix:(46)$$\left[\begin{array}{l} H_{k+t}^{v}\\ H_{k+t}^{g} \end{array}\right]\Phi_{k+t,k}=\left[\begin{array}{llllll} 0_{4\times 3} & 0_{4\times 3} & 0_{4\times 3} & \Pi^{\prime } & 0_{4\times 1} & 0_{4\times 3}\\ h_{k+t}^{1} & h_{k+t}^{1}(t_{k+t-1}-t_{k-1}) & h_{k+t}^{1}\Phi_{13}^{k+t} & 0_{3\times 5} & h_{k+t}^{2} & h_{k+t}^{3} \end{array}\right]$$
where
(47)$$\Pi^{\prime }=\left[\begin{array}{cc} \frac{\partial r_{C}}{\partial \delta \mu_{k+t}} & 0_{2\times 4}\\ 0_{2\times 1} & \frac{\partial r_{L}}{\partial \delta o_{k+t}} \end{array}\right]$$According to Equation (46), we can clearly observe that the dimension of the null space of M is equal to zero, which means the states represented in the ECEF frame are fully observable and the tightly coupled GNSS-VIO state estimator is consistent. □

The fact that the tightly coupled GNSS-VIO system is fully observable means that even if the system runs in long trajectories, the accumulated error can be eliminated. By leveraging the global measurements from GNSS, our system can achieve high-precision and drift-free positioning compared with the VIO system, which has four unobservable directions.

## 3. System Overview

The architecture of our proposed system is shown in Figure 4.

The proposed system implements four threads including data input, preprocessing, initialization, and nonlinear optimization. As shown in Figure 4, the white block diagrams represent the work that has been implemented by VINS-Mono [16] and GVINS [21], and the green block diagrams represent improvements we made. The inputs of our system are image, IMU, and GNSS measurements. In the preprocessing step, point and line feature detection and tracking were performed, IMU measurements were preintegrated, and the pseudorange measurements were smoothed by the carrier phase. In the initialization step, we followed the routine of VINS-Mono [16] for VI-Alignment. After VI-Alignment was completed, we performed GNSS initialization, which was divided into four stages: ENU Origin Estimation (Coarse), Yaw Estimation, ENU Origin Estimation (Fine), and GNSS-IMU Calibration. The first three stages were implemented in GVINS [21]. Finally, nonlinear optimization was performed. Nonlinear optimization will optimize the states in Equation (17) by leveraging the residuals and Jacobian matrices of different factors, which were computed in Section 2.5.

## 4. Experimental Results

Our experiments were conducted on the public dataset GVINS-Dataset [29], which captured scenes from the Hong Kong University of Science and Technology. The measurements were collected by a helmet that is equipped with a VI-Sensor and a u-blox ZED-F9P GNSS receiver. The dataset *sports field* captured a sports field scene where the device followed an athletic track for five laps. The sports field is an outdoor environment with an open area where the satellites are well locked and the RTK solution remains fixed. The other dataset *complex environment* was a complex indoor–outdoor environment where many challenging scenes were captured. For example, point and line features cannot be detected in bright or dim scenes, and the GNSS signal was highly corrupted or unavailable in cluttered or indoor environments (about 25 m). The overall distance of the *complex environment* dataset was over 3 km. We compared our proposed system with some open-source SLAM systems, including GVINS and VINS-Mono where VINS-Mono includes results with and without loop closure. For comparison with our results, we transformed the trajectories of VINS-Mono from the local world frame to the ECEF frame. Furthermore, since RTK has centimeter-level positioning precision, we compared it with our trajectories as the ground truth.

### 4.1. Sports Field

We plotted the trajectories of the sports field on Google Earth as shown in Figure 5. We see that VINS-Mono without loop closure suffers from accumulated drift among all three directions, which leads to the worst positioning precision among methods. Obviously, the drift increases with each lap around the sports field. VINS-Mono with loop closure can significantly improve the positioning precision, but there is still an obvious drift in yaw direction, which is mainly because the VINS-Mono has four unobservable directions. As a comparison, since GVINS is fully observable in the ECEF frame, it has a smaller positioning error and is drift-free. As depicted in Figure 5, its trajectory is very close to RTK. In addition, since our method is also a tightly coupled GNSS-VIO system, which is fully observable in the ECEF frame and the ENU frame, the positioning result of our method is also very accurate.

To quantitatively analyze results, we compared the positioning errors of GVINS and our method in detail. As shown in Figure 6, we visualized the positioning errors of GVINS and our proposed method. We see that the results of GVINS and our system fluctuate within a small range in the ENU frame and ECEF frame. The positioning errors of our method are less than 1m among all three directions of the ENU frame. Compared with GVINS, the error of our system is smaller in the east and north directions. In addition, the error of our method is smooth and stable in the ECEF frame, which is because we smooth the pseudorange measurements with the carrier phase. In addition, we listed the RMSE and Spherical Error Probable (SEP) for the positioning error of GVINS and our method in the ENU frame and ECEF frame. From Table 1, we can see that the RMSE of our positioning errors is lower than that of GVINS except in the up direction. The SEP refers to the radius of a sphere in which 50% of the estimated positions occur, and it can also evaluate the positioning precision of our system. The smaller the SEP, the higher the positioning precision of our system. We see that the SEP of our method in the ECEF frame is lower than that of GVINS, which shows that the positioning precision of our method is more accurate in the ECEF frame.

### 4.2. Complex Environment

In the following, we conducted experiments on the dataset *complex environment*, and we compared VINS-Mono, GVINS, RTK, and our proposed method. As shown in Figure 7, since VINS-Mono has four unobservable directions, the trajectory of VINS-Mono has a large drift. In addition, we saw that a section of the trajectory of VINS-Mono deviated from RTK by about 34 m, which is an unacceptable result. On the contrary, since GVINS and our proposed system are tightly coupled GNSS-VIO systems, there is no accumulated drift theoretically. In addition, due to a section of the trajectory being collected indoors, the GNSS receiver cannot obtain measurements at this time, which causes the failure of the RTK solution. In contrast, even if the GNSS measurements cannot be obtained, our system works well with VIO, and our trajectory is still accurate.

To further compare the performance of GVINS and our method, we analyzed the positioning errors of the two methods in the dataset *complex environment* as shown in Figure 8. It should be noted that we compared the RTK solution with our method as the ground truth. However, when the receiver is occluded, the satellite signal cannot be received, and the RTK solution is useless at this time. Thus, we only compared with segments where the RTK solution is available. Due to the complexity of the dataset, the overall positioning precision of the system is inferior to that of the sports field. The RMSE and SEP of GVINS and our method are shown in Table 2. We see that the positioning errors of GVINS in the east and north directions are slightly lower than that of our method, while the positioning error of our method in the up direction is much lower than that of GVINS. The SEP metric also shows that our method outperforms GVINS on this dataset.

Furthermore, since the ENU frame is gravity-aligned, we further analyzed the 2D trajectory error in the East–North plane as shown in Figure 9. The trajectory projected on the East–North plane is stained in red. The *z*-axis represents the positioning errors in the corresponding East–North plane, and the errors are represented by a scatter plot. In some bright or dim scenes, only a few point and line features can be detected, which will affect the positioning precision of the system. In addition, buildings, indoor environments, and trees will block satellite signals, which also degrade the performance of our system. Therefore, the positioning error in the complex environment is larger than that in the sports field, and the maximum error exceeds 2 m. As depicted in Figure 9, a sudden increase in positioning error occurs at the curves of the trajectory due to the large change in the orientation direction. On the contrary, the positioning error of the trajectory with small curvature is smoother compared with that of the trajectory with large curvature, which shows that our method is still effective even in complex scenes.

### 4.3. Observability Analysis

In Section 2.7, we demonstrated that the states represented in the ECEF are fully observable. However, according to [20], if the states are represented in the local world frame, the GNSS-VIO system has still four unobservable directions. Therefore, we can perform observability analysis from experimental results. Figure 10 shows the 3D trajectories of the complex environment represented in the ECEF frame and local world frame. We zoomed in on the trajectories of turning and climbing stairs for the convenience of the analysis. From Figure 10b, we see that since the states are unobservable in the local world frame, the two trajectories of climbing the same stairs have a drift. Even if the GNSS measurements are included in our system, the two trajectories are still inconsistent. In contrast, due to the states represented in the ECEF frame being fully observable, the two trajectories of climbing the same stairs are consistent and very close. Similarly, the same conclusions can be obtained for the trajectories of turning as shown in Figure 10. According to the results of the observability analysis, there is an obvious drift in the two trajectories of turning in the local world frame, while the phenomenon does not occur in the ECEF frame. In addition, the consistency of the system also shows that even in large-scale scenarios, it can still achieve high-precision positioning results as illustrated in Figure 10a.

### 4.4. Smoothed Interval

Equation (15) requires the receiver to continuously lock the carrier phase. If the receiver loses lock, the smoothed pseudorange is corrupted by any cycle slip that occurs and must be reinitialized when that happens. Therefore, it is necessary to investigate the influence of different smoothed intervals on the positioning precision of our system. The positioning error with six different settings is illustrated in Figure 11. When the smoothed interval is set to 200 epochs, the system has the largest positioning error in the three directions of the ENU frame. This happens because the receiver loses lock frequently within a smoothed interval and then cycle slip occurs, which causes an increase in the positioning error. By reducing the smoothed interval, the phenomenon that the receiver loses lock can be eliminated, which can improve the positioning precision of the carrier phase smoothed pseudorange. Obviously, with the smoothed intervals of 5, 10, 20, and 50 epochs, our system has smaller errors in east and north directions compared to the smoothed intervals of 100 or 200 epochs. However, if the smoothed interval is too small, the positioning precision in up direction cannot be further improved, and may even become worse. As shown in Figure 11, when the smoothed interval is set to 5 epochs, the positioning error in the up direction is the largest among different settings. The reason for this phenomenon is that when the smoothed interval is too small, the carrier phase cannot smooth the pseudorange well, which causes the smoothed pseudorange measurement to degenerate into the pseudorange measurement.

### 4.5. Line Feature Tracking Threshold

In the nonlinear optimization stage, if the number of keyframes for which line features are continuously observed is less than a threshold, then these line features are filtered out. A different threshold has a great impact on the robustness and positioning precision. As shown in Figure 12, a threshold of 2 means that those line segments that are observed by two consecutive keyframes or more can be added to the factor graph. However, since these line segments are continuously tracked for too few frames, they are not stable enough for nonlinear optimization, which will definitely reduce the positioning precision. On the contrary, if the line feature tracking threshold varies between 3 and 4, the proposed system can obtain more accurate positioning results in the ENU frame. Because the line segments observed by three or four consecutive keyframes are more stable and more suitable for nonlinear optimization.

## 5. Conclusions

In this paper, we propose a tightly coupled GNSS-VIO system based on point-line features. First, since line features contain more environmental structure information, we added line features to the system. Second, the pseudorange measurement can only achieve meter-level positioning precision but is fast and unambiguous. On the contrary, the carrier phase measurement can achieve the centimeter-level positioning precision but is required to solve the whole-cycle ambiguity, which is time-consuming. Therefore, we propose to combine the advantages of the two measurements and replace the pseudorange with the carrier phase smoothed pseudorange, which can greatly improve the performance of our system. Third, we considered the extrinsic translation parameter between the GNSS receiver and IMU, and our system can perform real-time parameter calibration. Finally, we demonstrated that if the states are represented in the ECEF frame, the tightly coupled GNSS-VIO system is fully observable. We conducted experiments on public datasets, which show our system achieves high-precision, robust, and real-time localization. In the future, we will further focus on improved methods for tightly coupled GNSS-VIO systems.

## Figures and Tables

**Figure 1 sensors-22-03391-f001:**
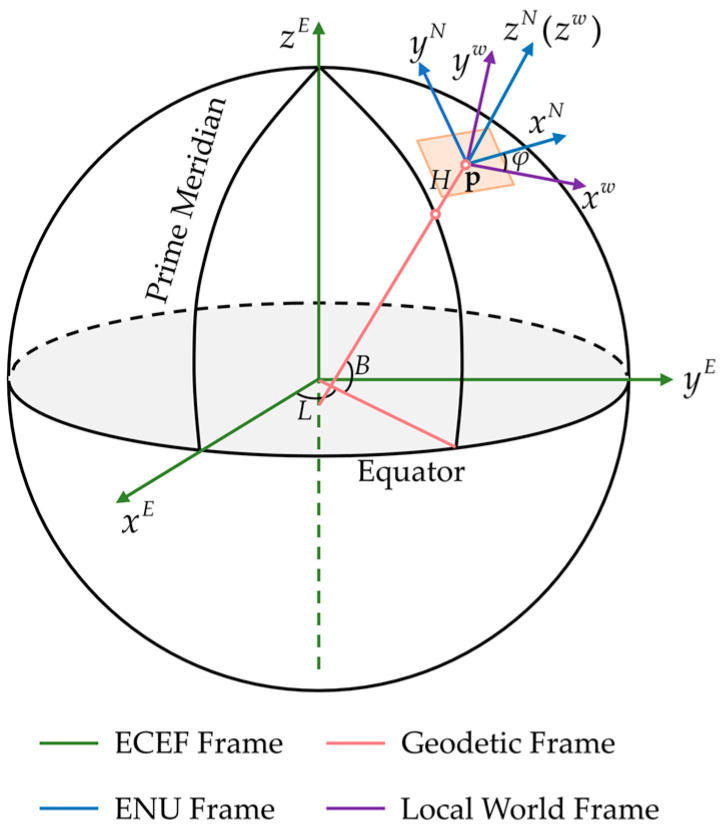
An illustration of the ECEF, Geodetic, ENU, and local world frames.

**Figure 2 sensors-22-03391-f002:**
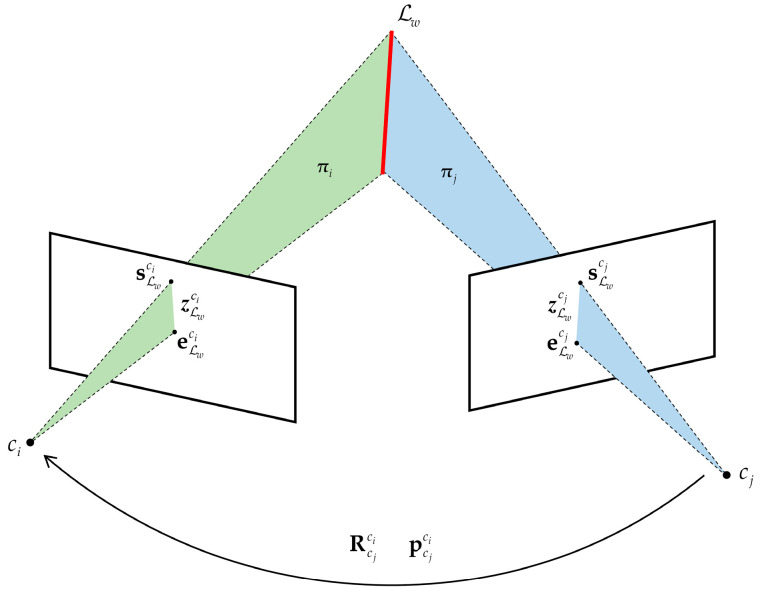
Line feature triangulation.

**Figure 3 sensors-22-03391-f003:**
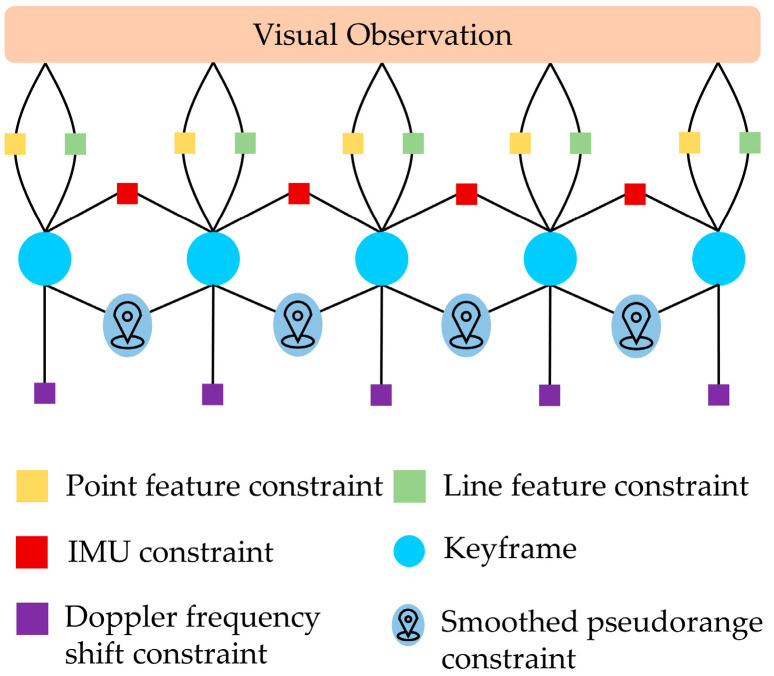
Factor graph illustration of our system.

**Figure 4 sensors-22-03391-f004:**
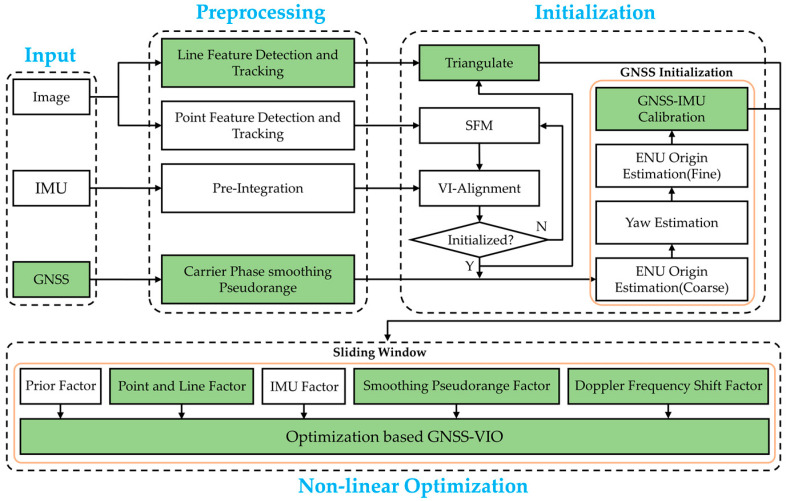
A block diagram illustrating the workflow of our proposed system.

**Figure 5 sensors-22-03391-f005:**
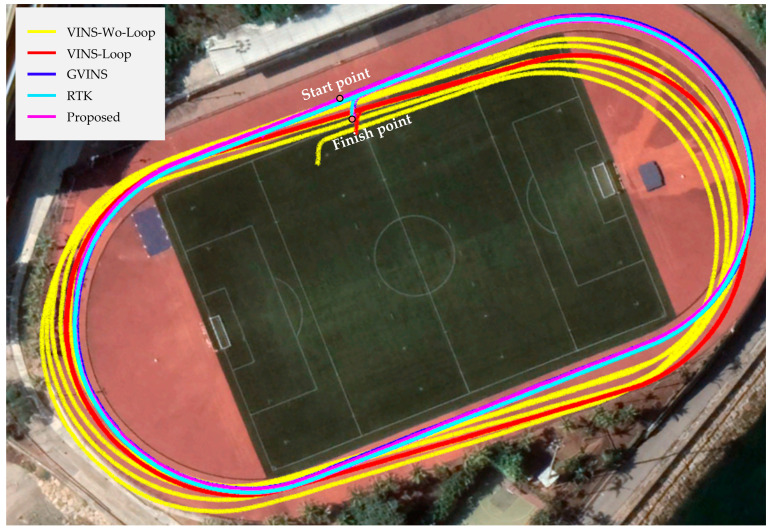
The trajectories of VINS-Mono, GVINS, RTK, and our proposed method in the sports field.

**Figure 6 sensors-22-03391-f006:**
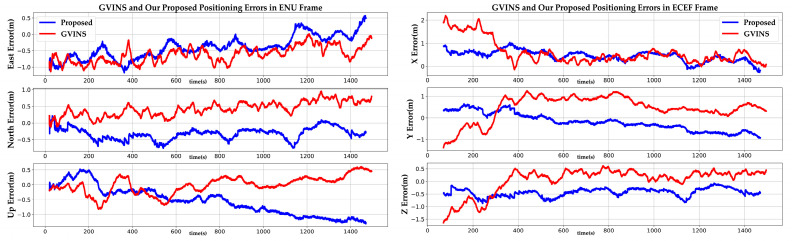
The positioning errors of GVINS and our proposed system in the sports field. The left three rows are the positioning errors in the east, north, and up directions of the ENU frame. The right three rows are the positioning errors in the X, Y, and Z directions of the ECEF frame.

**Figure 7 sensors-22-03391-f007:**
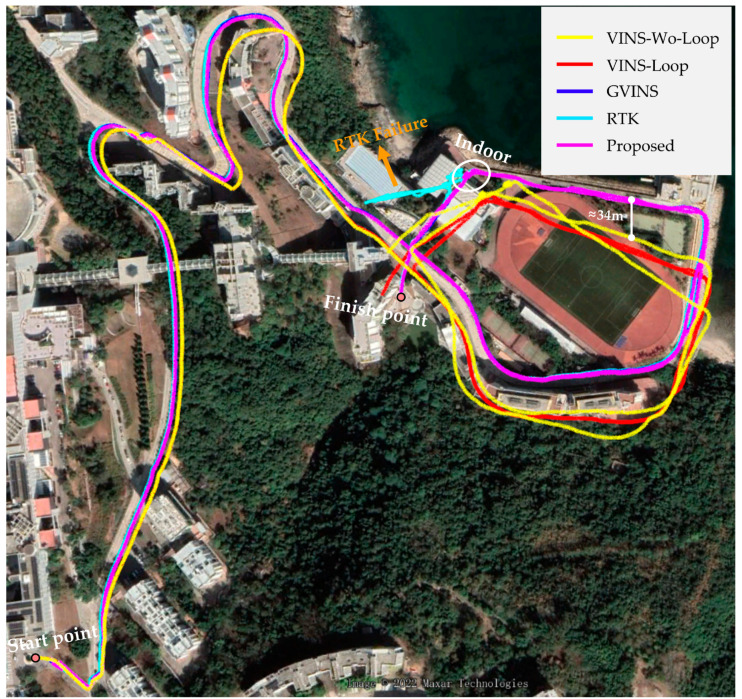
The trajectories of VINS-Mono, GVINS, RTK, and our proposed method in complex environments.

**Figure 8 sensors-22-03391-f008:**
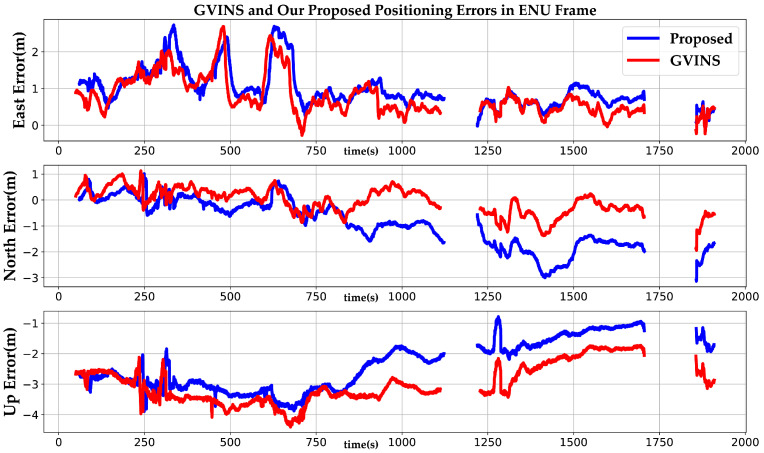
The positioning errors of GVINS and our proposed system in the complex environment.

**Figure 9 sensors-22-03391-f009:**
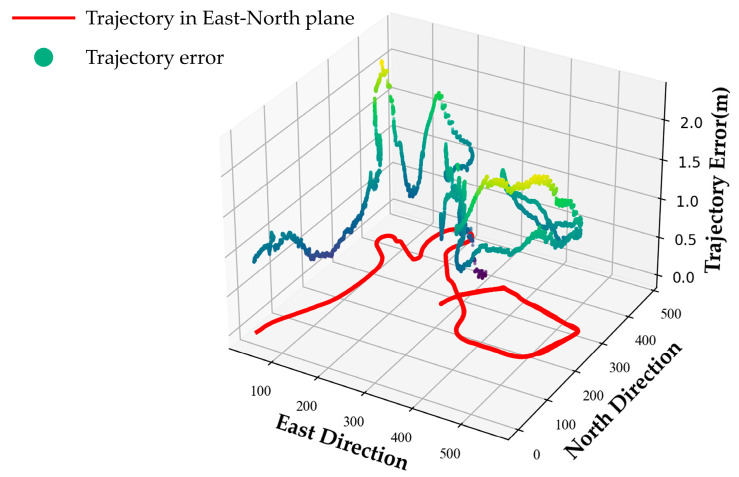
The 2D trajectory error of the complex environment on the East–North plane.

**Figure 10 sensors-22-03391-f010:**
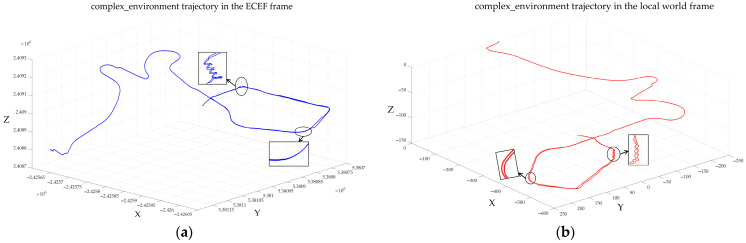
Comparison of the 3D trajectories represented in different frames. (**a**) is the ECEF frame and (**b**) is the local world frame.

**Figure 11 sensors-22-03391-f011:**
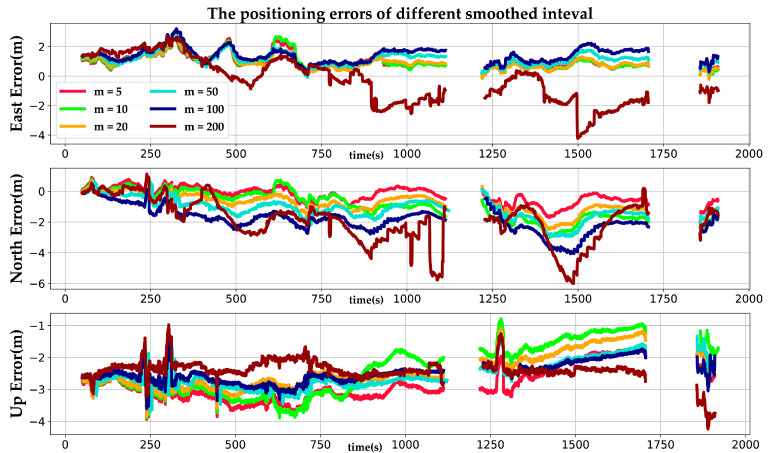
The positioning errors with different smoothed interval settings (*m* = 5, 10, 20, 50, 100, and 200).

**Figure 12 sensors-22-03391-f012:**
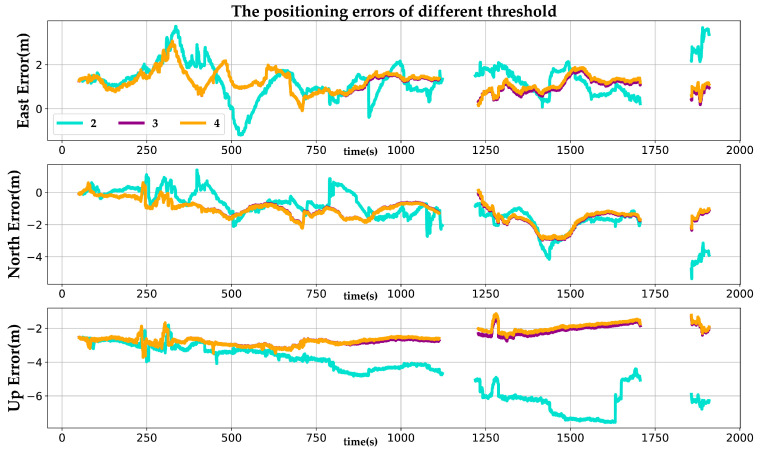
The positioning errors of different line feature tracking thresholds.

**Table 1 sensors-22-03391-t001:** RMSE[m] and SEP[m] statistic of GVINS and our proposed method in the sports field.

	ENU Frame				ECEF Frame			
	RMSE			SEP	RMSE			SEP
	East	North	Up		X	Y	Z	
GVINS	0.661	0.477	**0.296**	**0.683**	0.834	0.766	0.581	1.002
Proposed	**0.524**	**0.360**	0.725	0.761	**0.522**	**0.440**	**0.466**	**0.651**

**Table 2 sensors-22-03391-t002:** RMSE[m] and SEP[m] statistic of GVINS and our proposed method in the complex environment.

	ENU Frame			
	RMSE			SEP
	East	North	Up	
GVINS	1.091	0.782	3.016	2.540
Proposed	1.230	1.404	**2.499**	**2.390**

## Data Availability

Not applicable.

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
