# Peer review of "A Tightly Coupled Visual-Inertial GNSS State Estimator Based on Point-Line Feature"

_sensors, 2022, doi:10.3390/s22093391_

Round 1

Reviewer 1 Report

I have read this submission with great interest, and  overall, this is well written, however, I think the followings need to be addressed for acceptance.

(1) p.3 “We propose a new method to calibrate the extrinsic translation parameter between GNSS receiver and IMU in real-time.”

If the calibration method is one of the contributions, please clarify what is the novelty of the method, and provide comparative evaluation experiments with other calibration methods to demonstrate the effectiveness. If those are difficult, remove the calibration method from descriptions about contributions.

(2) p.12 “To facilitate the observability analysis of our proposed system, some simplifications are required”

Please add any disadvantages or limitations regarding simplification.

(3) p.12 “Theorem 1”

Without any introduction, Theorem 1 appears suddenly and its proof begins. First, please explain the position of the theorem in this paper, and then describe the theorem and its proof.

(4) Positioning errors in 4. Experimental Results

The positioning technology community often uses CEP (Circular Error Probable) for 2D and SEP (Spherical Error Probable) for 3D as a metric for evaluating positioning error. They are also already an ISO standard. So please add SEP or CEP to the results of each experiment as much as possible.

Cf. ISO/IEC 18305:2016, Information technology — Real time locating systems — Test and evaluation of localization and tracking systems

(5) Figure 7

Please specify which sections of the route were indoor, and describe the distance and passing time of the indoor section.

(6) 4.4. Smoothed interval

It reads as if simply shortening the interval will solve the problem. If there are any disadvantages or limitations to shortening the interval, please mention them.

In addition, the followings are editorial comments for revision:

(1) Figures

Maybe due to image compression, the image quality of each figure is not good, and some of the text in the figures is difficult to read. Please make the text in the rasterized figures easier to read by increasing the font size or by other means.

(2) p.4 “v^{w}_b, b_{a}, and b_{a} denote”

The second b_{a} seems to be b_{g}.

(3) p.7 "In front of the wave source, the wavelength becomes shorter and the frequency becomes higher due to the compression of the wave. On the contrary, in the back of the wave source, the wavelength becomes longer and the frequency becomes lower."

Is that correct? What do you assume the relative motion between them and its direction? What is front and what is back?

(4) Figures 5 and 7

Please indicate the start and finish points in the figures.

(5) Figure 10

The stair area should be expanded more. Also, it is unclear from what perspective it is possible to see if they are drifting or not.

Author Response

We thank the reviewers’ comments concerning our manuscript. Those comments are valuable and very helpful. We have read through comments carefully and have made corrections. Please see the attachment.

Reviewer 2 Report

The paper presented tightly coupled GNSS–Visual–Inertial odometry system based  on point-line features. Authors also propose a method to calibrate the extrinsic translation parameter between GNSS receiver and IMU.

The overall structure of the paper is fine. Title is appropriate and matches the content of the paper. The abstract is clear, organized and well written. The introduction provides sufficient background and includes relevant references. The experiment is well designed, and the results look sound and correct. Even though some corrections need to be addressed throughout the paper, the English is acceptable, and the ideas are clearly transmitted to the reader. Although old references are cited in the text, the manuscript also contain citations to the state-of-the-art.

Additional comments:

  • Please, provide the fill reference of paper 21: S. Cao, X. Lu and S. Shen, "GVINS: Tightly Coupled GNSS–Visual–Inertial Fusion for Smooth and Consistent State Estimation," in IEEE Transactions on Robotics, doi: 10.1109/TRO.2021.3133730.
  • Why presenting the acronim SOTA if it is never used in the document?
  • Consider increasing the resolution of all Figures.
  • Figures 6, 11, and 12 are not readable. Their font is too small, and they get blurry when zoomed in. Please, consider increasing the image size.
  • Although the introduction discusses that in weak textures and dark scenes the pose estimation of point-line-based visual odometry can be significantly affected, the manuscript lacks a deeper discussion on how the proposed system would behave in unstructured environments (where low-level visual cues such as lines traditionally fail)? Are the IMU measurements sufficient to cope with a low number of point and line features over multiple visual observations? How would the proposed system benchmark in such environments?

Author Response

(The authors gave the same response as above.)
